



# Firn microstructure variations under extreme metamorphism at Allan Hills Blue Ice Area, Antarctica

B. Ilyse Horlings[1], Annika N. Horlings[2], Kaitlin M. Keegan[3], Julia Marks Peterson[4], and Zoe Courville[1,5]

[1]Thayer School of Engineering, Dartmouth College, Hanover, NH 03755, USA
[2]Institute of Arctic and Alpine Research, University of Colorado Boulder, Boulder, CO, 80303, USA
[3]Department of Geological Sciences and Engineering, University of Nevada, Reno, NV 89557, USA
[4]College of Earth, Ocean, and Atmospheric Sciences, Oregon State University, Corvallis, OR 97331, USA
[5]Cold Regions Research and Engineering Laboratory, Hanover, NH 03755, USA

**Correspondence:** B. Ilyse Horlings (ilyse.horlings@dartmouth.edu)

**Abstract.** The spatial variability of firn properties, including microstructure, is critical for mass-balance estimates and ice-core interpretations. This includes local and regional variations in firn microstructure, which influence compaction and air movement in the firn. Some of the oldest ice cores are located in blue-ice areas and originated from low-accumulation areas. Surface processes, including small variations in surface conditions and topography, may cause significant alteration of the firn

microstructure and bulk firn properties in low-accumulation areas, as firn may interact with surface processes for long periods of time (e.g., hundreds of years). Here, we examine the impact of very low-accumulation rates ($A = \mathcal{O}(10^{-3} - 10^{-2}$ m a$^{-1}$)) on firn microstructure using micro-computed tomography data from a firn core (PICO2303) retrieved in the accumulation zone at Allan Hills Blue Ice Area (BIA), Antarctica during the 2023-2024 NSF-COLDEX field season. First, we analyze microstructural parameters such as total porosity, density, specific surface area, degree of anisotropy, and structure thickness,

as well as 2D and 3D reconstructed digital datasets. Subsequently, we compare our observations with a firn core from Dadic et al. (2015) that we refer to here as AHMIF, and located ∼9 km downslope. In the PICO2303 core, we find 1) faceted grains, 2) wind crusts, 3) vertically oriented networks, and 4) specific surface area of 1.5 m$^2$ kg$^{-1}$ greater and density of 205 kg m$^{-3}$ lower below the surface compared to those measured in the AHMIF core. We propose that these characteristics are a result of differences in both surface processes (e.g., formation and decay of sastrugi) and subsurface evolution (e.g., depth-

hoar formation) that likely occur at varying degrees. We model surface snow density as a function of wind speed, which is approximated from a katabatic wind model. We find modeled surface density closely predict data at AHMIF and PICO2303 with errors of 8.2% and 0.4%, respectively, which suggests that core differences are influenced by spatial variability of surface wind and topography. Broadly, the differences between AHMIF and PICO2303 may impact near-surface air movement with resultant implications on post-depositional modification of stable water-isotope ratios and gas exchange.

## 20  1  Introduction

Firn is the transitional material between snow and glacial ice, and, by definition, is older than one year. Over time, firn becomes buried from accumulating snow at the surface, which causes compaction and metamorphism with depth due to a variety of




processes, including grain settling, recrystallization, and sintering. Because firn is porous and permeable, its compaction and
metamorphism generates a decrease in interstitial air space between surface snow grains to isolated bubbles at depth. Diffusion
of gases through the firn depends on the structure of the grain/air space, or microstructure, and occurs until there is negligible
permeability (e.g., Schwander and Stauffer, 1984; Sowers et al., 1992). Below the depth of negligible permeability, the gases
are entrapped and provide direct atmospheric records at the time the open porosity becomes closed. Additionally, surface and
near-surface processes can promote the exchange of molecules between the solid and vapor phases of water, and result in
modification of stable isotopologue ratios of water from those originally set during the formation of snow precipitation (e.g.,
Clarke and Waddington, 1991; Waddington et al., 1996; Albert, 2002; Waddington et al., 2002; Albert et al., 2004; Wahl et al.,
2022). Both processes require information about firn that is often estimated from firn models, which historically have used bulk
depth-density profiles and strain measurements for calibration (Herron and Langway, 1980; Arthern et al., 2010). However,
studies have found that density does not always dictate differences in permeability and gas diffusivity (e.g., Adolph and Albert,
2014; Gregory et al., 2014). Rather, evolving firn microstructure (e.g., specific surface area, ice-network density, anisotropy)
influences such bulk properties and can change the overall physical, thermodynamic, and geochemical systems of the firn
through time (e.g., Albert et al., 2000, 2004; Calonne et al., 2011, 2017, 2019).

Empirical models used to describe firn compaction and densification were derived from firn measurements taken from
sites with environmental conditions that fall close to the Clausius-Clapeyron (CC) relation (Fig. 1), linking temperature and
saturation vapor pressure. There are many polar sites of scientific interest with conditions that fall outside of this range of mean
annual temperature ($T$) and accumulation rate ($A$). Thus, it's important to investigate firn properties and compaction processes
at locations with conditions that are farther from the CC relation. Sites like the Allan Hills Blue Ice Area (BIA), Antarctica,
experience $T$ and $A$ that cannot be predicted by the CC relationship. The underlying blue ice at Allan Hills BIA is not derived
from the firn in the region and is under the influence of high katabatic winds, topographically-driven accumulation, and other
micro- and macro-topographical effects, and therefore offers a unique setting to understand properties and dynamics in very
low-accumulation conditions. For example, from Figure 1, temperatures at Allan Hills BIA are comparable to those at Summit,
Greenland and WAIS Divide, Antarctica, however, accumulation rates are several orders of magnitude lower than these sites.

Understanding the influence of $T$ and $A$ on firn microstructure and properties is important for interpreting the range of
resultant observations across the $T$-$A$ space. However, even local differences may exist at an individual site, especially for low-
accumulation ones. Studies such as Courville et al. (2007) have shown that horizontal spatial variations in low accumulation
rates (<40 mm weq yr$^{-1}$) result in large differences in firn microstructure and properties over 3.4 km, influencing vertical spatial
variations within the firn layer and amplifying the effect of other conditions (e.g., wind, temperature gradient) (Courville et al.,
2007).

Studies of firn metamorphism and microstructure in very low-accumulation regions exist only from a few sites in East
Antarctica. For example, Courville et al. (2007) and Albert et al. (2004) focused on the megadunes, a region where accumula-
tion varies by an order of magnitude across low-amplitude, long-wavelength features (0-5 mm weq yr$^{-1}$ versus 20-30 mm weq
yr$^{-1}$)(see Figure 1). Calonne et al. (2017) investigated another low-accumulation site (8.1 mm yr$^{-1}$) at Point Barnola near Dome
C of Central East Antarctica (see Figure 1). These kinds of studies inform us of several main structural characteristics in firn



**Figure 1.** The temperature and accumulation at previous ice-core sites (black dots) and the Clausius-Clapeyron relationship (blue line). Future core sites proposed at Taylor Dome are in green, and Allan Hills BIA is in orange. Sites with similar temperature but varying accumulation are outlined in light red boxes; those with similar accumulations but varying temperature are outlined in light yellow boxes.





in low-accumulation regions. At the snow-atmosphere surface, high winds often drive high initial densities from impact-snow deposition into a wind-packed surface that has lower permeability than the underlying firn due to post-depositional processes

(Albert and Shultz, 2002; Albert and Rick, 2004). Within the top several meters of firn, well-sintered and mono-sized grains may evolve from faceted depth hoar and create high permeability (e.g., Rick and Albert, 2004; Albert et al., 2004; Courville et al., 2007). While mechanisms such as grain settling and boundary sliding are traditionally assumed to be controllers of firn compaction at shallow depths in high-accumulation, low-wind regions, they may be partially or completely inactive at low-accumulation, high-wind sites (e.g., Dadic et al., 2015; Calonne et al., 2017). These conditions have been observed to

generate near-constant densities with depth in near-surface firn, especially in East Antarctica (e.g., Endo and Fujiwara, 1973; Dadic et al., 2015; Weinhart et al., 2020; Inoue et al., 2024). Thus, firn near the surface may begin closer to the closest packing density yet not experience grain settling and boundary sliding, and, as a result, provides a deviation from theory (e.g., Herron and Langway, 1980) and a recipe for developing unique sets of microstructure.

Additionally, in near-surface firn, temperature-gradient metamorphism occurs through vapor diffusion and transport (e.g., the

sublimation of warmer grains to colder grains) (Inoue et al., 2024). In high-accumulation areas, cold air over warm snow near the surface generates vertical vapor transport, and generally produces low-density, coarse-grained depth hoar formed during late summer and autumn (Alley, 1988). Strong winds deposit fine-grained high-density wind crusts and slabs at the top of the depth hoar that may be punctuated with ice crusts from condensation, summer insolation, and high temperatures (Alley, 1988). With prolonged exposure to the atmosphere, the distinction between the stratification of depth hoar, wind crusts/slabs, and any

ice crusts may be smoothed (e.g., Calonne et al., 2017). Therefore, firn that has developed from very low-accumulation rates and consequently from extreme metamorphism undergoes a different transformation than other sites with higher accumulation rates. Near-surface firn may be exposed to surface conditions for long periods of time (e.g., hundreds of years), which can produce larger depth hoar crystals and more well-sintered grains at greater depths (Dadic et al., 2015). However, depth hoar formation may be suppressed if wind-packed surface densities are greater than 350 kg m$^{-3}$, which may occur at high-impact

depositional sites (e.g., Alley, 1988). In summary, a spectrum of firn microstructure may be observed from different local and climatic conditions.

Here, we investigate local spatial variability of firn microstructure in the accumulation region of Allan Hills BIA by (1) examining the microstructure of PICO2303, a firn core extracted during the NSF Center for Oldest Ice Exploration (COLDEX) 2023-34 austral field season, using micro-computed tomography, and (2) comparing our dataset with the microstructure dataset

from the AHMIF core extracted 9 km grid northwest and downslope in 2011 (Dadic et al., 2015). This research provides a foundation for understanding local variations in firn microstructure at very low-accumulation sites generated by discrete storm events from wind and topography effects similar to Allan Hills BIA.

## 1.1  Core Location

We compare two firn cores that have been retrieved from the Allan Hills BIA region (Figure 2). The first core was described in

Dadic et al. (2015) and was collected from a snowpatch within the blue ice field called the Allan Hills Main Ice Field (AHMIF) in January 2011 at -76.667, 159.25 (Figure 2). Within this region, sublimation is a dominant process but small topographical





depressions and low ice velocities (around 0.16 m a$^{-1}$) allow accumulation to occur for firn to develop (Dadic et al., 2015). A second core, PICO2303, was recovered during the austral summer of 2023-2024 by the NSF Center for Oldest Ice Exploration (COLDEX) around 9 km upglacier from the 2011 core, located in an area of topographically driven accumulation at -76.751,

159.11 (Figure 2). Half of the core was used for chemistry and the other half was cut into 10-cm samples. Every other 10-cm sample was scanned for micro-computed tomography measurements to around 11.3-m depth, as described below, and the other half archived.

## 1.2   Climate at Allan Hills BIA

The region of Allan Hills BIA is characterized by high winds and very low to negative accumulation of snow, and is a mixed area

of blue ice and snow and firn. The snow and firn is not the source of the blue ice and most likely has developed from intermittent accumulation of discrete events between periods of no or negative accumulation. The one estimate of the accumulation rate in the accumulation zone adjacent to the blue-ice area is 0.0075 m yr$^{-1}$ (Dadic et al., 2015). Near the PICO2303 core site, the wind speed, wind direction, and temperature data were recorded during summer 2010, 2016, 2019, and 2022 at a weather station (Figure 4). In this location, the measured wind speed across the summer seasons averages around 8 m s$^{-1}$ and may be as

high as 19 m s$^{-1}$ with a fairly consistent direction from the grid north. Daily temperature variations may exceed 5°C, and can deviate up to 15°C within a time span of a month during the summer.

## 1.3   Micro-computed tomography (micro-CT)

Imaging of the firn core is performed with a Skyscan micro-computer tomographer with a resolution of 39.49 $\mu$m and the following scanner settings: voltage=40 kV, current=200$\mu$A, exposure=300 ms, and rotation step=0.6°. Processing of the micro-

CT images includes filtering using a Gaussian blur, thresholding using the automatic (Otsu) method, and despeckling to remove small groups of pixels that are from error (e.g., Otsu et al., 1975; Hagenmuller et al., 2016; Courville et al., 2019; Li and Baker, 2021). Once completed, CTAn, a proprietary software package used for processing micro-CT data, can then perform 2D and 3D analyses to estimate the selected parameters describing microstructure, e.g., total porosity, density, degree of anisotropy (DA), specific surface area (SSA), and structure model index.

Here, total porosity is defined as the ratio of pore volume to total volume for both open and closed pores,

$$\phi_{\text{tot}} = \frac{\text{Po.V(tot)}}{\text{TV}} \times 100\%, \tag{1}$$

where Po.V(tot) is the volume of all open and closed pores, and TV is the total volume of firn. Density is calculated through the following micro-CT parameters,

$$\rho_{\text{firn}} = \frac{\frac{\text{Obj.V}}{\text{TV}}}{\rho_{\text{ice}}}, \tag{2}$$

where Obj.V is the volume of ice and $\rho_{\text{ice}}$ is the density of ice (917 kg m$^{-3}$). Degree of anisotropy (DA) is a parameter to assess isotropy, the presence or absence of preferential alignment of structures along a directional axis. DA ranges from 0 to 1 for

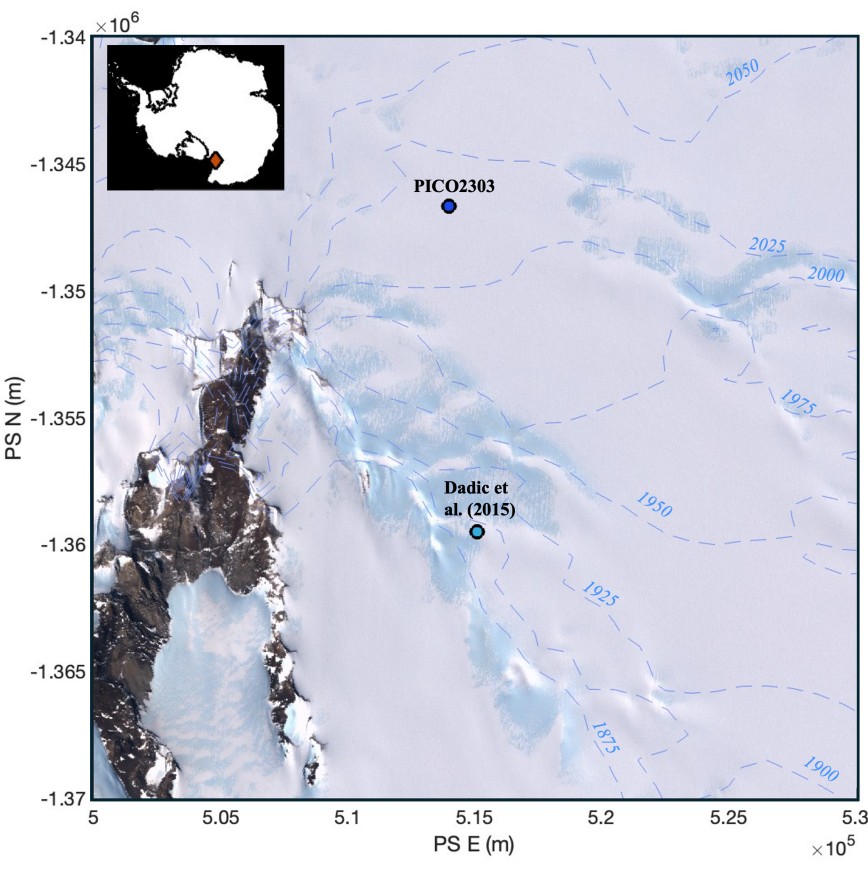

**Figure 2.** A map showing location of the two firn cores discussed in this paper at Allan Hills BIA, Antarctica: PICO2303 firn core (dark blue circle) and the firn core collected by Dadic et al. (2015) (light blue circle). Elevation contours from REMA (Howat and others, 2019) are superimposed on Natural-Color, Pan-Sharpened imagery that combines Landsat ETM+ bands 3,2,1 (red, green, and blue of the electromagnetic spectrum) from the Landsat Image Mosaic of Antarctica (LIMA) Projection. Blue shaded color on the map represents the wind-swept glacial blue ice that outcrops at the surface. The map inset shows location of figure domain in Antarctica. The projection is polar stereographic (EPSG: 3031).



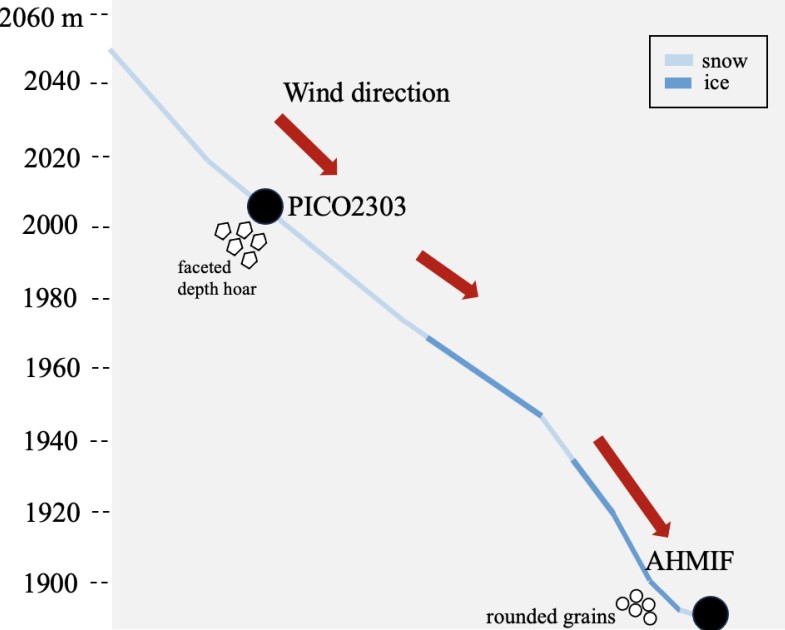

**Figure 3.** Schematic elevation profile and wind of sampling sites, PICO2303 and AHMIF. Wind is influenced on several spatial scales by the geometry of the snow-air interface, both by macro and micro-topography. The schematic highlights the former. The line color highlights the areas of topographically-driven accumulation on the surface inferred from the map of Figure 4.2. Site PICO2303 is approximately 9 km from the AHMIF.

total isotropy and total anisotropy, respectively. It is calculated from the minimum and maximum eigenvalues from the eigen analysis of a second-order, orthogonal tensor describing an anisotropy ellipsoid fitted over the mean intercept lengths through the 3D image volume such that,

$$\text{DA} = \left[1 - \left(\frac{\lambda_{\min}}{\lambda_{\max}}\right)\right],$$ (3)

where $\lambda_{\min}$ is the smallest eigenvalue and $\lambda_{\max}$ is the largest eignevalue. Specific surface area (SSA), in $m^2$ $kg^{-1}$, is defined to follow Dadic et al. (2015) as the ratio of the air-ice interface area to ice mass,

$$\text{SSA} = \left[\frac{\text{Obj.S}}{\text{Obj.V}}\right]\left[\frac{1}{\rho_{\text{ice}}}\right],$$ (4)

where Obj.S is the object (ice) surface ($mm^{-2}$) and Obj.V is the object volume ($mm^3$). Structure thickness, in mm, indicates the
largest sphere diameter encircling a point in the ice matrix and completely bound within the solid surface. For firn specifically, it is a measure of the characteristic ice particle size where a particle may be one or a number of crystals and grains (e.g., Hildebrand and Rüegsegger, 1997; Li and Baker, 2021). The average of a parameter is calculated as the mean across images (e.g., $\rho_{\text{av}} = (\rho_1 + \rho_2 + ... \rho_{\text{n}})/n$), where n is the number of images and $\rho$ is the parameter of interest.



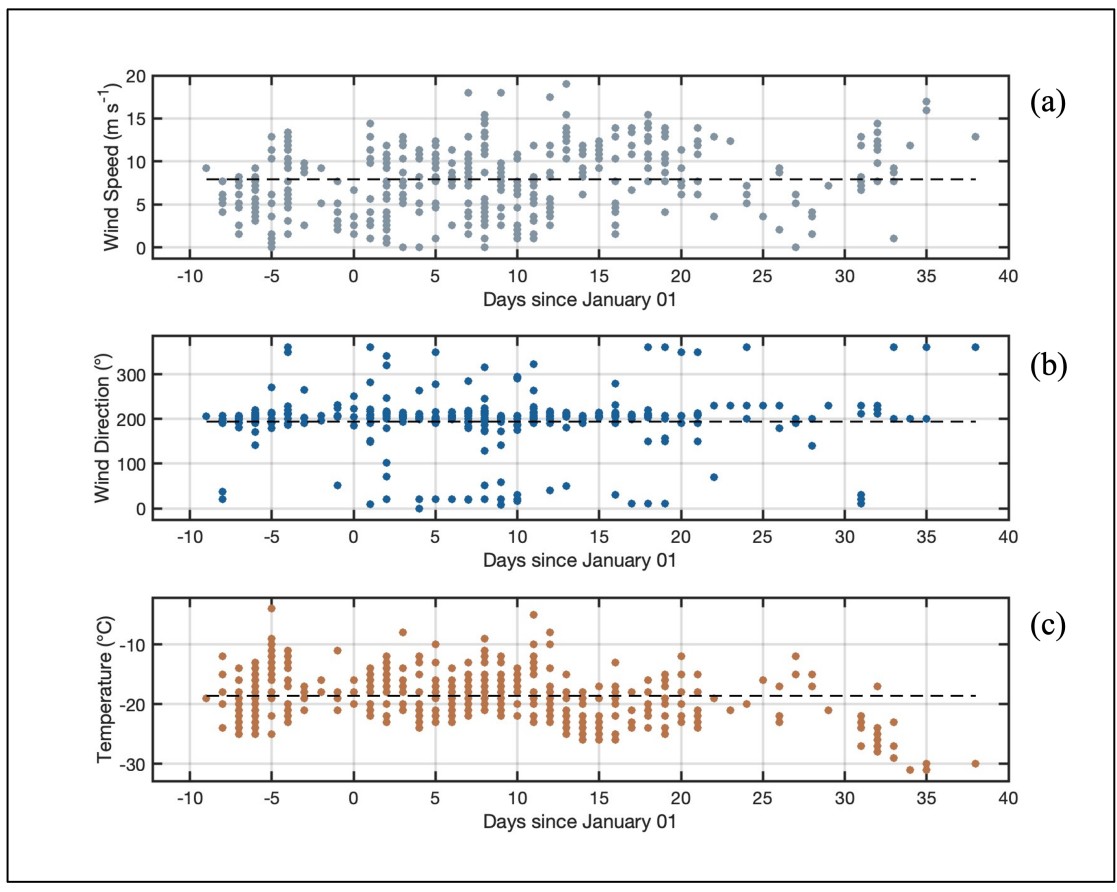

**Figure 4.** Wind speed relative to grid north (a), wind direction (b), and temperature (c) data from the weather station from summer of 2010, 2016, 2019, and 2022 (Naval Information Warfare Center Atlantic, Polar Programs Integrated Product Team (NPP), 2024). Black dashed lines are the averages across all data.

Through micro-CT analysis, we also identify other features. For example, the micro-CT scanner initially generates 2D
projection images where the vertical structure of each firn sample may be observed. From these images, we can identify wind crusts as small (<3 mm), oblique or horizontal lineations that appear dark in the raw image data because of their higher density.

## 1.4 Comparison of cores

We compare micro-CT data from both cores. Micro-CT data from the AHMIF core is described in Dadic et al. (2015) and based on eight firn samples taken from 0.61 - 4.69 m. Each sample had a diameter of 36 mm and was scanned with a resolution
of 36 $\mu$m. From a shallow snow pit, cast samples were retrieved at a site within 50 m of the AHMIF core location (Dadic et al., 2015). The PICO2303 core sits higher in the accumulation zone while the AHMIF core site lies closer to the blue ice with an





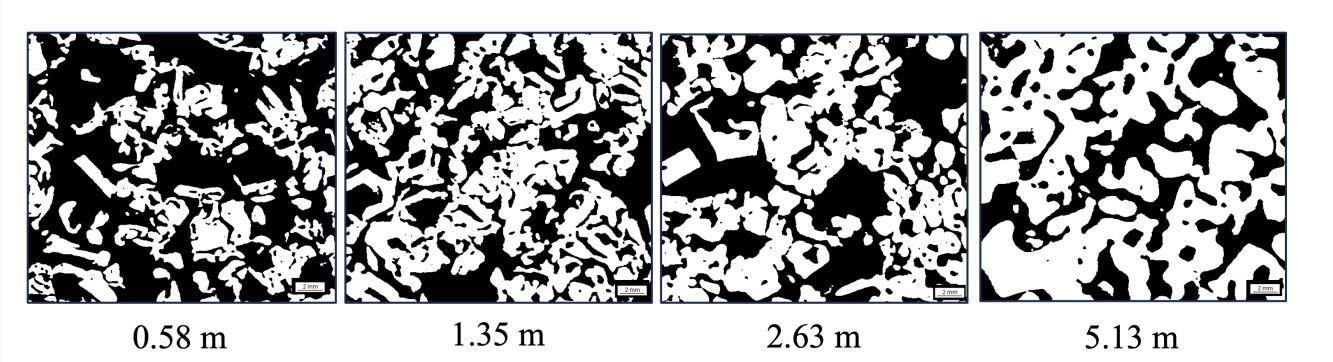

| 0.58 m | 1.35 m | 2.63 m | 5.13 m |

**Figure 5.** Examples of binary images showing the evolution of faceted grains in the PICO2303 core. White is ice and black is void space. From left to right: 0.58 m, 1.35 m, 2.63, 5.13 m.

elevation difference of around 100 m (Figure 2). We compare the parameters (density, SSA, structure thickness) and features of the AHMIF core from Dadic et al. (2015) to those of the PICO2303 core.

## 2 Results

We observe several characteristics in the PICO2303 core: (1) a fine-grained layer above 0.2 m; (2) vertically oriented faceted depth hoar beginning at 0.45 m depth; (3) larger faceted crystal pockets among smaller faceted, higher-density networks of depth hoar below 0.45 m; (4) identifiable wind crusts at 0.02 m, 0.25 m, and 0.58 m depth (Figure 6); (5) SSA that decreases from the snow-atmosphere interface ($>7.6$ m$^2$ kg$^{-1}$) to lower values at 4.7 m (2.8 m$^2$ kg$^{-1}$) and 11.3 m (1.68 m$^2$ kg$^{-1}$) (Figures 7 and 8); and (6) densities that do not substantially change with depth from the near surface of 0 - 0.5 m ($\rho_{av} = 387$ kg m$^{-3}$) to 0.5 - 5 m ($\rho_{av} = 365$ kg m$^{-3}$) (Figures 7 and 8). The AHMIF core shows similar trends in (5) - (6) with (A) high near-surface SSA (5.17 - 34.78 m$^2$ kg$^{-1}$) that decrease until 4.7 m (1.6 m$^2$ kg$^{-1}$) (Figure 8); and (B) densities that do not show substantial change from 0.5 - 5 m ($\rho_{av} = 570$ kg m$^{-3}$) (Figure 8); however, in the AHMIF core, there is (C) no observable depth hoar, (D) no observable wind crusts, and (E) homogeneous, rounded grain networks (Figure 9). This is in contrast to our observation in the PICO2303 core. Lastly, the AHMIF core generally has (F) higher SSA from 0 - 0.5 m (SSA$_{av}$ = 12.54 m$^2$ kg$^{-1}$) and SSA that, on average, is 1.5 m$^2$ kg$^{-1}$ lower from 0.5 - 5 m (Figure 8); and (G) higher density ($\Delta\rho_{av} = 72$ kg m$^{-3}$) from 0 - 0.5 m and ($\Delta\rho_{av} = 205$ kg m$^{-3}$) from 0.5 - 5 m (Figure 8).

We describe the structure of each core in detail. The fine-grained layer in the PICO2303 core in the upper 0.2 m transitions into a layer of large, faceted grains of depth hoar (Figure 5a). Several small wind crusts lie within the fine-grained layer in the upper 0.2 m. First, we observe an oblique wind crust at 0.02 m (Figure 6) that progresses from right to left in images (1, 2, 3) from shallower to deeper in the sample. Second, we observe higher porosity ($\phi_{av} = 69.44\%$) below the wind crust (left) compared to above (right), where we observe lower porosity ($\phi_{av} = 61.54\%$) and a more interconnected network. Third, we also see a horizontal wind crust at 0.25 m (Figure 6, panel 2) and at 0.56 m (Figure 6, panel 3). The latter is the thickest wind crust



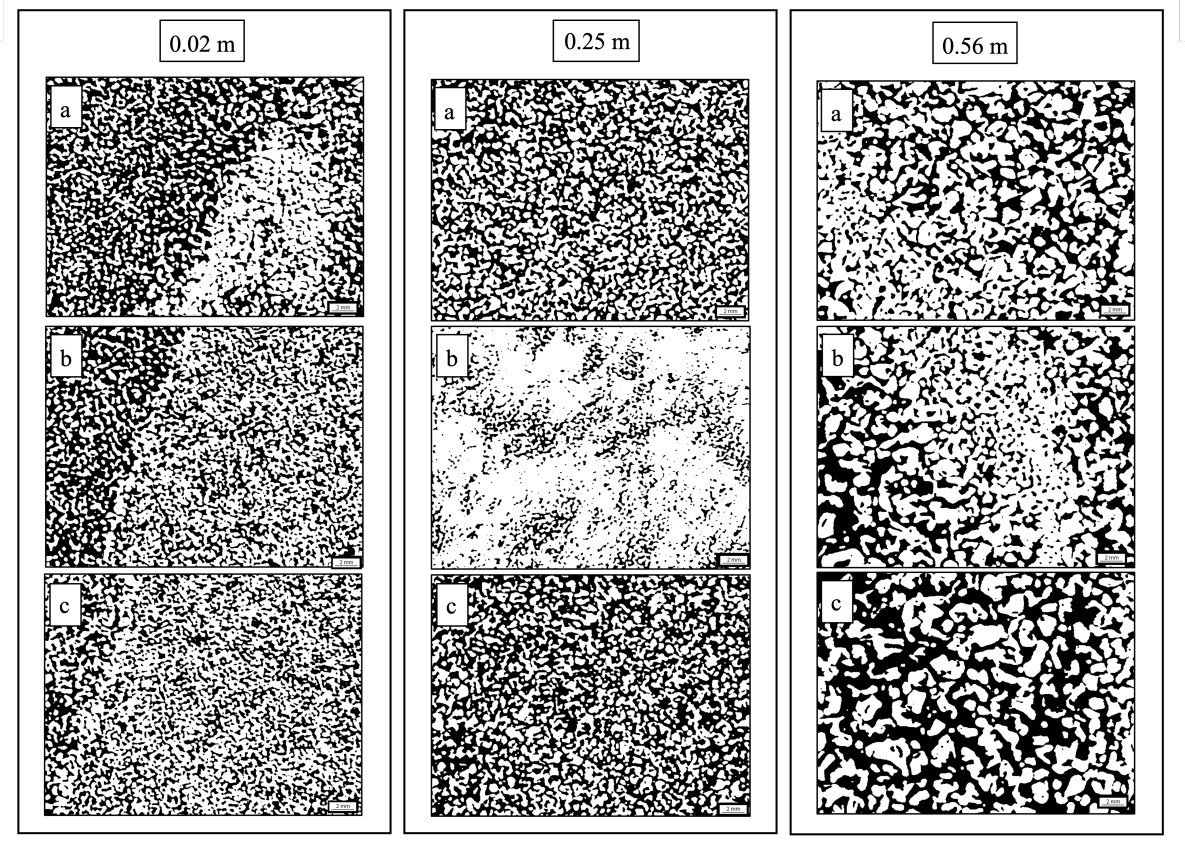

**Figure 6.** Panels of binary images showing wind crusts at (left to right) 0.02 m, 0.25 m, and 0.56 m. White is ice and black is void space. Top to bottom (shallower to deeper in the core) images within each panel represents above (a), within (b), and below (c) the wind crust.

with an average vertical height of 3 mm. Both low-porosity wind crusts at 0.25 m and 0.56 m are surrounded by high-porosity firn. Finally, we observe pockets of large facets and low porosity, and more rounded facets and high porosity.

We use the micro-CT data to derive the parameters listed in section 2.3 (see Figure 7) to describe their variations across the full depth of PICO2303 core. Porosity decreases from 0 - 1 m ($\phi_{av} = 63.2\%$) to 11.3 m ($\phi = 44.8\%$). Density follows the inverse of porosity and increases with increasing depth to 522 kg m$^{-3}$ at 11.3 m. Porosity and density trends correspond in part to the evolution of the depth hoar with depth (Figure 5). The observed faceted grains surrounded by large void spaces, for example, image at 0.58 m of Figure 5, are indicative of the shift to lower density and higher porosity seen above and at 0.58 m (Figure

7). Additionally, degree of anisotropy (DA) does not show a clear trend (see Figure 7), and is instead characterized by large fluctuations from around 0.15 - 0.4 (i.e. larger values mean higher degree of anisotropy) throughout the core. SSA decreases with increasing depth from 7.6 - 9.3 m$^2$ kg$^{-2}$ above 0.5 m to 1.7 m$^2$ kg$^{-1}$ at 11.3 m, where the rate of change of SSA decreases with depth. Finally, structure thickness increases linearly from around 0.40 mm at the surface to 2.1 mm at 11.3 m depth, where the rate of change increases at around 3.5 m.



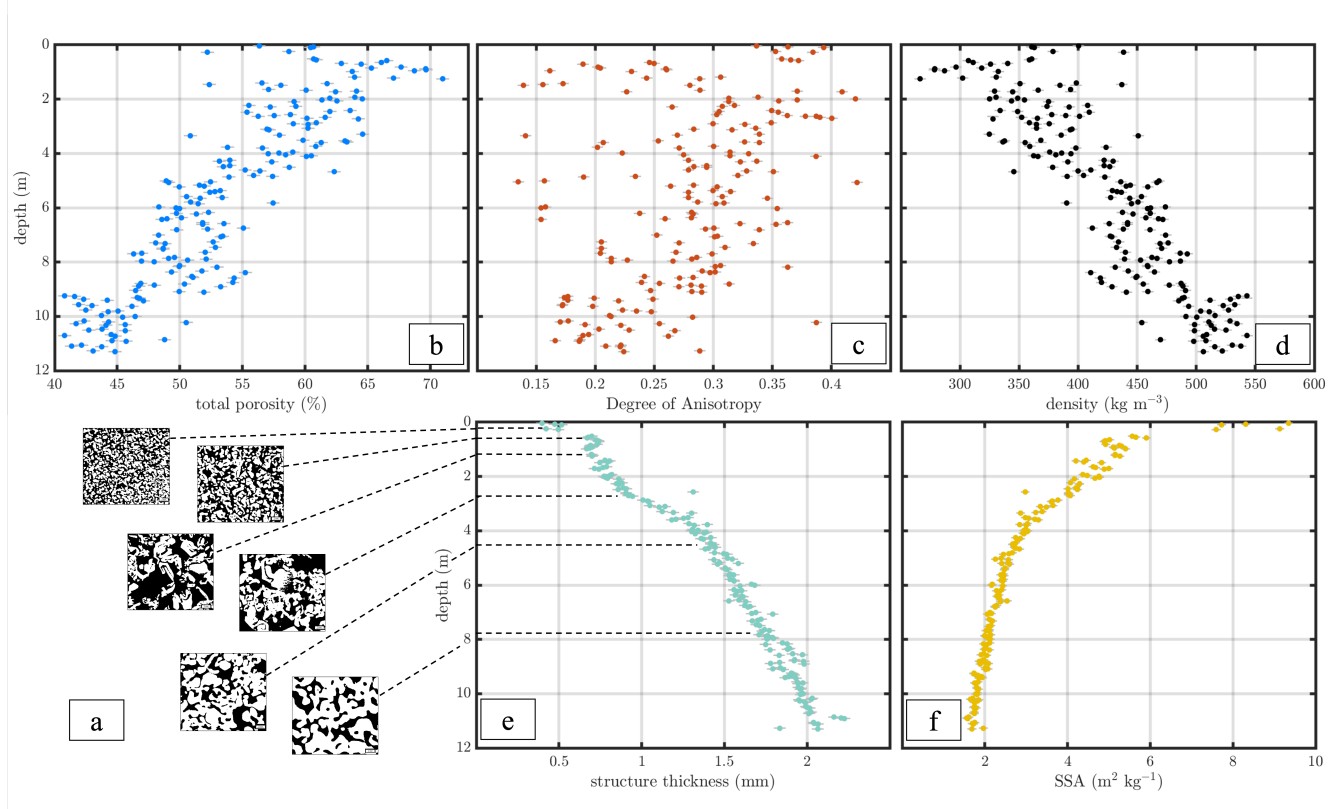

**Figure 7.** Binary images of microstructure at various depths from the firn core where black represents void space and white represents firn grains (a). Microstructure parameters with depth: total porosity (b), degree of anisotropy (c), density (d), structure thickness (e), and SSA (f). Standard errors are shown by gray bars.

## 3 Discussion

### 3.1 Operating processes

We suggest that the following processes contribute to the observed stratigraphy and microstructure in the PICO2303 core. First, a fine-grained layer commonly represents a wind-packed structure and signifies the spatial redistribution of snow (Gow, 1965; Calonne et al., 2017). Second, wind crusts are thin (<2 mm), high-density layers that form originally at the surface due to post-depositional wind drift and wind packing, and can become preserved within the firn (Weinhart et al., 2021; Li et al., 2023). Further, the oblique wind crusts at 0.02 m and 0.56 m indicate that, at the time of formation, the snow surface was inclined on a slope, which contrasts the flat surface required to create the wind crust at 0.25 m. The mixture of wind-crust geometry also indicates that the snow surface has fluctuated and the firn column is non-monotonic with time. Third, the presence of depth hoar indicates that temperature gradients and low densities near the surface existed during its formation with surface densities >350 kg m$^{-3}$ (e.g., Sommerfeld, 1983; Li et al., 2023). Fourth, combinations of pockets of large, faceted grains





**Figure 8.** Comparison of parameters between the PICO2303 and AHMIF core: structure thickness (a), specific surface area (SSA) (b), and density (c).




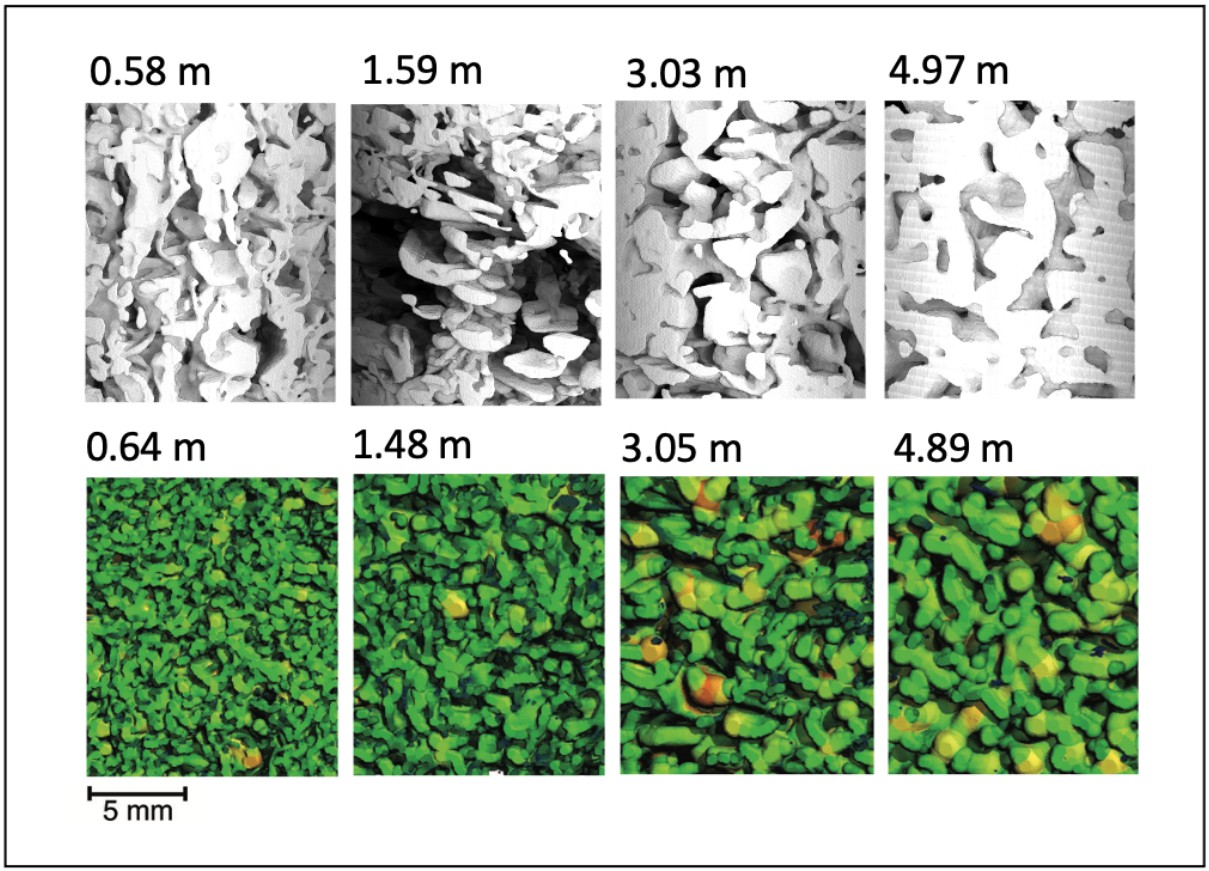

**Figure 9.** Comparison of 3D reconstructions of Fig. 4 of AHMIF from Dadic et al. (2015) with the PICO2303 core. Note that average parameter values are taken per scan, which is dependent on how the core was cut; thus, exact depth comparisons are not performed here.

among smaller faceted grains may suggest sastrugi remnants (Calonne et al., 2017). The lack of distinct layering is indicative of long and sustained formation of snow and firn from low-accumulation rates and exposure to the effects of wind and other snow-atmosphere conditions (e.g., Courville et al., 2007).

We suggest that the distinct variations in structures between the PICO2303 and AHMIF core may be driven by differences
in macro- and micro-slope geometry as well as wind conditions (e.g., speed, scouring) at the snow-air interface. In part, higher wind speeds at the AHMIF core site would drive higher surface SSAs and densities, which, in turn, would suppress depth hoar formation if above a certain density as well as possibly above a certain wind speed. The lack of evidence of wind crusts in the AHMIF core may also be driven by wind scouring, which would disburse the snow before it could accumulate.

Within the firn, the microstructure and post-depositional metamorphic processes may create feedbacks, as microstructure
and surface processes co-evolve. Wind crusts may restrict parts of the air column, and steep temperature gradients often create high-porosity layers below large-grained wind crusts, where the wind crust itself may absorb the upward transport of




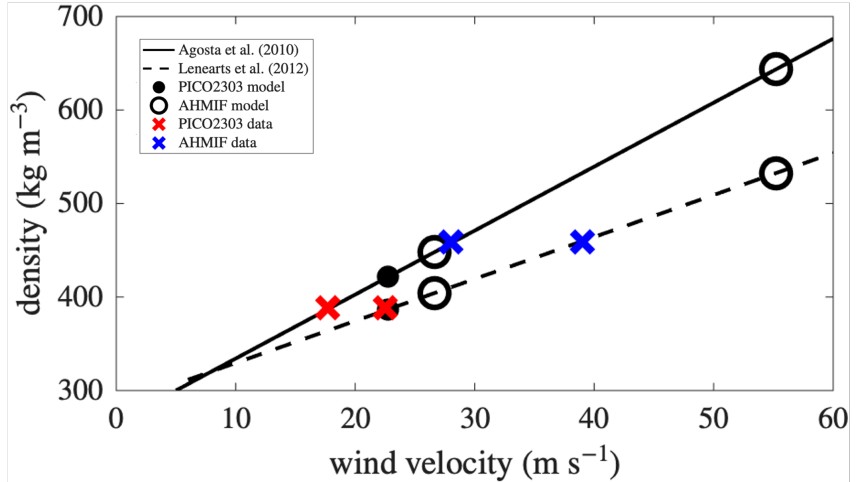

**Figure 10.** Modeled surface snow density from Lenaerts et al. (2012) and Agosta et al. (2019) for AHMIF and PICO2303. The red crosses represent the PICO2303 averaged surface data (0-0.25 m) while the blue crosses represent the averaged AHMIF surface data (0-0.35 m)

mass (e.g., Weinhart et al., 2021). Spatial variations in grain size and porosity may cause different recrystallization rates (i.e. rates are higher in large-faceted grain structures and lower in dense, small-grain structures), and enhance layer differences and stratigraphy. Highly anisotropic pockets of firn are caused by a preferred and heterogeneous direction of air movement,

which may become accentuated through preferential vertical or horizontal vapor transport and recrystallization processes as described above. Further, the presence of wind crusts, anisotropic grain networks, and low-porosity pockets of depth hoar in the PICO2303 core may increase or decrease air movement and permeability locally. Further, the high near-surface SSA and the shift to lower values at around 0.25 - 0.5 m in both the PICO2303 and AHMIF cores has been observed at other sites, such as Dome Fuji, and is a function of the accumulation rate as well as wind conditions (e.g., Domine et al., 2009; Linow

et al., 2012; Inoue et al., 2024). We suggest that post-depositional processes may not fully erase the imprint of initial surface conditions and near-surface structures (i.e. wind crusts), as seen by the sustained differences with depth between AHMIF and PICO2303 values in both SSA and density (see Figure 8). The persistence or eradication of wind crusts as well as the observed microstructure difference indicate that the conditions at the snow-atmosphere surface (e.g., wind, ambient temperature, solar radiation) are integral in initiating the feedback of mechanisms and microstructure within the firn. We suggest that they may

also be an important control on resultant microstructure at local scales at sites where there are significant local variations in topography at the snow-air interface.

## 3.2 Effect of wind

The two core sites are geospatially located 9 km apart in different local areas of the very low-accumulation area of Allan Hills BIA. The site of the PICO2303 core is located in the accumulation area at an elevation of around 2000 m, with a

variable firn depth that can reach up to 35 m (Conway, 2017). The AHMIF core site is located downslope in a snow patch



underlain by firn in the blue-ice area at around 1900 m, with an unknown firn depth that is most likely extremely shallow because of its adjacency to the blue-ice area. In the Allan Hills BIA accumulation region, differences in microstructure are most likely impacted by differences and consistency in topographical and surface-roughness features, wind speed, ambient and firn temperature, solar radiation, their interplay, and the influence of these environmental variables on near-surface ventilation, the temperature gradient, and vapor exchange (e.g., Albert, 2002). We suggest that the spatial variation of wind speed along with micro- and macro-topographical differences in the geometry of the snow-air interface at the two sites are drivers of the differences we observe in AHMIF and PICO2303 cores (e.g., presence or absence of wind crusts, depth hoar, and associated microstructure variations).

To begin exploring the first-order influence of surface conditions, we obtain a first-order approximation to understand the local spatial variability in wind speed. Steep terrain enhances seasonally variable surface katabatic wind patterns (e.g., Spaulding et al., 2012) where wind acceleration is generally downslope and towards the blue ice, which both removes snow and ablates ice (e.g., Sugden and Hall, 2020). We consider the katabatic wind model used in Van Den Broeke and Bintanja (1995) for stationary flow in which we calculate speeds at the surface and also at a height of 10 m. Details of these calculations are presented below in Appendix 2.

Many surface snow density ($\rho_\mathrm{s}$) parameterizations exist as a function of wind among other variables (e.g., Kaspers et al., 2004; Lenaerts et al., 2012; Vionnet et al., 2012; Gallée et al., 2013; Agosta et al., 2019). We use Lenaerts et al. (2012) and Agosta et al. (2019), which incorporate mean surface temperature, $T$, and wind speed at 10 m, $u_\mathrm{s10}$, to estimate $\rho_\mathrm{s}$ at AHMIF and PICO2303, defined as

$$\rho_\mathrm{s} = 97.5 + 4.49 u_\mathrm{s10} + 0.77T \tag{5}$$

$$\rho_\mathrm{s} = 149.2 + 6.84 u_\mathrm{s10} + 0.48T, \tag{6}$$

respectively. The estimated $\rho_\mathrm{s}$ are shown in Figure 10. The modeled $\rho_\mathrm{s}$ for Lenaerts et al. (2012) (LN) and Agosta et al. (2019) (AG) fall close to the values of the surface density data for both AHMIF and PICO2303 (Figure 8). The lower AG estimation (421 kg m$^{-3}$) for AHMIF shows an 8.2% error between the averaged surface density data from 0 - 0.35 m (459 kg m$^{-3}$) while the LN estimation (387 kg m$^{-3}$) for PICO2303 shows a 0.4% error between the averaged surface density data from 0 - 0.25 m (388 kg m$^{-3}$). Overall, wind and the interrelationship with macro-topography influences surface density in these areas, and we conclude through first-order approximation that wind speed may be one factor in causing the structural (e.g., wind crusts) and microstructural (e.g., SSA, density) differences observed in the two cores.

In summary, microstructures (SSA and density) spatially vary between AHMIF and PICO2303. Depth-hoar and wind-crust formation also spatially varies, and is driven by macro- and micro-slope geometry as well as wind conditions (e.g., speed, scouring). Depth hoar at PICO2303 generates lower density and higher SSA below the snow-atmosphere interface while higher wind speeds generate high-impact deposition, and thus the higher density cap and higher SSA near the surface. On the micro- or local scale, wind crusts and depth hoar may impact air movement, either restricting or enhancing air flow



and causing preferential vapor transport. Variability in air movement across small spatial scales may impact gas and stable
water-isotope ratios for ice-core interpretation. Post-depositional alteration may occur at the surface from vapor exchange
with the atmosphere or mechanical processes such as wind redistribution and mixing of surface snow. Alteration may also
occur in the near-surface snow and firn from vapor transport within the firn column from mixing from turbulence driven by
pressure changes, or by advection from steady ventilation driven by pressure gradients over surface topography. Changes in
air movement from variability in firn structure and microstructure may change post-depositional alteration even on micro- or
local scales, notably at very low-accumulation areas, such as Allan Hills BIA, in which firn experiences long exposure to the
surface and near-surface processes as well as their feedbacks. Constraining variability in both firn structure and microstructure
measurements are needed near future ice-core drilling effort. This is especially true for low-accumulation areas in which the
firn is the origin of underlying ice (e.g., 2 - 8 cm a$^{-1}$ sites across Taylor Dome). Future work should investigate the contributions
of other surface and near-surface conditions, such as the effect of other macro- as well as micro-topography of the snow-air
interface, and their interplay on the temperature gradient and vapor exchange. Finally, permeability should also be investigated
in the near surface firn for understanding air movement as well as stable water-isotope ratios .

## 4 Conclusions

We use micro-computed tomography (micro-CT) to analyze the microstructure of a firn core, PICO2303, retrieved at Allan
Hills Blue Ice Area (BIA), Antarctica in 2023, and compare micro-CT data with the AHMIF core extracted 9 km downslope
in 2011. This dataset provides the first firn microstructure comparison at Allan Hills BIA and one of few micro-CT datasets
in a very low-accumulation region ($A = \mathcal{O}(10^{-3} - 10^{-2}$ m a$^{-1}$)) undergoing extreme firn metamorphism. We observe depth
hoar, wind crusts, and vertically oriented networks in the PICO2303 core which are not present in the AHMIF core. Further,
we observe lower densities ($\Delta\rho = 200$ kg m$^{-3}$) as well as higher SSA ($\Delta$SSA = 1.5 m$^2$ kg$^{-1}$) from 0.5 m - 5 m in the PICO2303
core in contrast to the AHMIF core. We suggest that these distinctions are a result of spatial variability of surface topography
and wind speed, and may be influenced by other interrelated snow-atmosphere variables (e.g., ambient temperature, solar
radiation). To assess the first-order approximation of the impact of wind speed on surface density, we approximate wind speed
at the two sites using a 1-D katabatic model, and feed our estimates into surface-density parameterizations. We find higher wind
speeds at AHMIF compared to PICO2303 which generate surface densities that agree well with averaged measured surface
densities of the cores (8.2% and 0.4%, respectively).

Firn at low-accumulation sites experiences long-term exposure to surface conditions as well as firn ventilation, which impacts
firn microstructure. Feedbacks of processes and microstructure may contribute to the spatial variability of stable water isotopes
ratios and gas diffusion through the firn, and impact the signal that is recorded and subsequent interpretation. Future work
should target exploring the spatial variation of microstructure and feedbacks in surface and post-depositional processes at low-
accumulation sites. It should also target obtaining more measurements of shallow firn cores across Allan Hills BIA and at other
low-accumulation regions.





*Data availability.* The processed 2-D and 3-D micro-CT parameter datasets are available at the U.S. Antarctic Program Data Center (USAP-DC): https://www.usap-dc.org/view/dataset/602002.

## Appendix A

Additional micro-CT reconstructions (see Figure A1).

## Appendix B

Within the model, upper air movement and advection are assumed to be negligible. Negative buoyancy from cooling air over a slope acts as the driving force within the model, which is balanced by both frictional and Coriolis forces such that

$$0 = -kVu + f\nu + g\frac{\Delta\theta}{\theta_0}\sin\alpha \tag{B1}$$

$$0 = -kV\nu - fu, \tag{B2}$$

where $k$ is a constant friction coefficient, $V$ is absolute wind speed, $u$ and $\nu$ are the wind components, $f$ is the Coriolis parameter, $g$ is gravity, $\Delta\theta$ is the inversion strength, $\theta_0$ is the background temperature, $\alpha$ is the angle of the surface slope. If equations B1 and B2 are rearranged for wind speed and direction, the following equations are produced,

$$V = \left[\frac{\cos(\delta)}{k}g\frac{\Delta\theta}{\theta_0}\sin(\alpha)\right]^{1/2} \tag{B3}$$


$$\cos(\delta) = \frac{f^2}{2k\frac{\Delta\theta}{\theta_0}\sin(\alpha)} + \left[\left(\frac{f^2}{2k\frac{\Delta\theta}{\theta_0}\sin(\alpha)}\right)^2 + 1\right]^{1/2}, \tag{B4}$$

where $\delta$ is the angle between the surface wind direction and downslope vector.

To use these equations at Allan Hills BIA, we prescribe $k = 1.25 \times 10^{-5}$ m$^{-1}$, which Van Den Broeke and Bintanja (1995) use and is based on measurements at Mizuho Station, which is a site also characterized by average katabatic wind speeds of 10

m s$^{-1}$ and a similar mean temperature of around -32°C (e.g., Yamanouchi et al., 1981; Watanabe et al., 1978). We then use REM at 5-m contour resolution to approximate the average slopes over 6 km around AHMIF, PICO2303, and the weather station sites. For AHMIF, the topography is complex and so the wind direction may or may not correspond to the weather station. Thus, we estimate a range of slopes from NNE to SE to generate a range of velocities. We tune $\Delta\theta$ to the weather station which has the recorded wind velocity and then use this value in equation B3 to estimate the wind speed at AHIMIF and PICO2303.

We find modeled wind speeds of 14.1 m s$^{-1}$ (PICO2303) and 16.4 - 34.2 m s$^{-1}$ (AHMIF) (see Figure B1). The highest winds recorded at PICO2303 ($\sim$15.4 m s$^{-1}$) seen in Figure 4 are within the range of wind speeds calculated for AHMIF.







**Figure A1.** 3D reconstructions of samples at 0.02 m, 0.45 m, 1.15 m, 2.38 m, 7.89 m, and 19.09 m (left to right and top to bottom) showing the progression from the upper fine-grained layer into the faceted depth hoar into the rounded network.





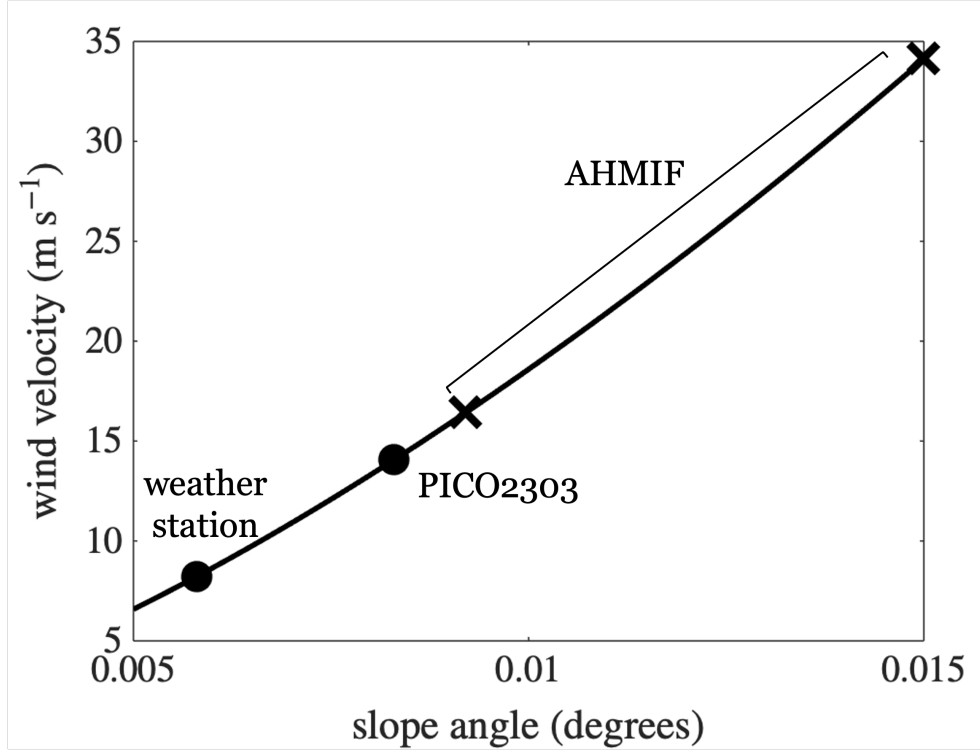

**Figure B1.** Modeled wind speeds at the weather station, PICO2303, and AHMIF. A range is give for AHMIF because of its more complex surrounding topography.

Many snow parameterization equations require $u_{s10}$, and we must estimate the vertical wind profile in the atmospheric surface layer. We use the definition referenced by Van Den Broeke and Bintanja (1995),

$$u_z = \frac{u^*}{\kappa} \left[ ln\left(\frac{z}{z_0}\right) + \psi\left(\frac{z}{L}\right) \right], \tag{B5}$$

where $u^*$ is the friction velocity, $z_0$ is the aerodynamic surface-roughness length, and $\psi\left(\frac{z}{L}\right)$ is a stability correction. Values of $u^*$, $z_0$, and $\psi\left(\frac{z}{L}\right)$ are averaged from Table 1 of Van Den Broeke and Bintanja (1995) and adjusted to generate surface wind velocities calculated at AHMIF and PICO2303, where we find the $u_{s10}$=17.0 m s⁻¹ and $u_{s10}$=20.1 - 41.3 m s⁻¹ for PICO2303 and AHMIF, respectively.

*Data availability.* The microstructure data is publicly available at https://www.usap-dc.org/view/dataset/602002. All outputs from this study
are also available from the authors without conditions.



*Author contributions.* B.I.H and A.N.H conceptualized the study. B.I.H. performed the micro-CT scanning, data analysis and interpretation, and writing of the manuscript. J.M.P. drilled the firn core. A.N.H. processed the firn cores, and Z.C. provided access to laboratory resources. A.N.H and K.M.K contributed to data analysis and interpretation. A.N.H contributed to data visualization as well as the review and editing of the manuscript. K.M.K, Z.C., and J.M.P. contributed to the editing of the manuscript.

*Competing interests.* At least one of the (co-)authors is a member of the editorial board of The Cryosphere.

*Acknowledgements.* This work was supported by the U.S. National Science Foundation Center for Oldest Ice Exploration (NSF COLDEX), an NSF Science and Technology Center (NSF Award #2019719), and "Collaborative Research: A New Approach to Firn Evolution using the Taylor Dome Natural Laboratory" (NSF Award #2024132). We thank the U.S. National Science Foundation Ice Core Facility (NSF-2041950) for ice core sampling assistance and curation. We thank the NSF Office of Polar Programs, the NSF Office of Integrative Activities,
and Oregon State University for financial and infrastructure support, and the NSF Antarctic Infrastructure and Logistics Program, the NSF Ice Drilling Program, and the Antarctic Support Contractor for logistical support. We thank the NSF Ice Drilling Program for support activities through NSF Cooperative Agreement 1836328.



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
