# Peer review of "Firn microstructure variations under extreme metamorphism at Allan Hills Blue Ice Area, Antarctica"

_EGUsphere, 2025_

## Referee Comment (RC1)

**Referee Report on**

**"Firn microstructure variations under extreme metamorphism at Allan Hills Blue Ice Area, Antarctica"**

by Horlings et al.
* * *
**General assessment**

The manuscript presents a valuable new micro-CT dataset from an extremely low-accumulation environment at Allan Hills Blue Ice Area (BIA), Antarctica. Such datasets are rare and of high relevance for understanding firn evolution under conditions relevant to Oldest Ice exploration. The observational work is carefully executed and the comparison with the AHMIF core (Dadic et al., 2015) is well motivated.

However, while the dataset itself is strong, the **interpretation is currently not sufficiently supported by quantitative analysis**, and several physical arguments are either incomplete or inconsistent. In its present form, the manuscript reads largely as a descriptive case study, while multiple causal interpretations are proposed without statistical validation.

I therefore recommend **major revisions** before the manuscript can be considered for publication in *The Cryosphere*.
* * *
**Major comments**

**1. Lack of statistical testing and hypothesis validation**

The manuscript proposes several hypotheses regarding the role of wind speed, surface topography, and firn exposure time in controlling microstructural differences between the PICO2303 and AHMIF cores. However, **no statistical tests are presented** to support these claims.

- Differences in SSA, density, and structure thickness are discussed qualitatively, but it remains unclear whether these differences are statistically significant relative to within-core variability.

- At minimum, variance estimates, confidence intervals, or formal hypothesis tests comparing the two cores should be provided.

- Without such analysis, the robustness of the inferred differences remains uncertain.

**2. Vertical spatial variability insufficiently quantified**

Although vertical profiles of microstructural parameters are shown, **vertical spatial variability is not explicitly analysed or discussed**.

- Depth-dependent trends, regime changes, or variability within specific layers (e.g. surface layer vs. depth-hoar-dominated firn) are not quantified.

- Given the importance of vertical firn structure for permeability, ventilation, and post-depositional processes, this represents a significant shortcoming.

- The authors should consider analysing vertical gradients, depth-dependent variance, or structural breakpoints.

**3. Alternative hypotheses not adequately considered**

The manuscript strongly emphasizes wind speed and surface topography as dominant controls, but **alternative explanations are not sufficiently discussed**, including:

- Differences in firn age (especially concerning the two cores) and residence time near the surface

- Temporal variability in accumulation and storm frequency

- Post-depositional vapor transport processes not directly linked to wind speed

A more balanced discussion explicitly addressing alternative hypotheses would strengthen the scientific interpretation and avoid over-attribution.

**4. Physical interpretation requires clarification and tightening**

Several physical processes are invoked in a way that lacks clarity or precision:

- Metamorphism, compaction, and porosity reduction are sometimes conflated.

- Depth hoar formation is described as being "suppressed" under certain conditions, which is physically misleading (see specific comments).

- The links between observed microstructure, air movement, permeability, and isotope modification remain largely speculative.

The manuscript would benefit from a clearer distinction between **observations**, **inferred mechanisms**, and **speculative implications**.

**5. Missing age constraints on firn layers**

The manuscript repeatedly refers to long exposure times (hundreds of years), yet **no age estimates are provided**.

- Even approximate age constraints (e.g. accumulation-based, order-of-magnitude estimates) would substantially strengthen the interpretation.

- Without age information, it is difficult to assess the persistence of microstructural features with depth or to compare the two cores in a physically meaningful way.

**6. Role of horizontal movement and advection**

Horizontal ice and firn movement is mentioned but not evaluated.

- Horizontal advection may influence stratigraphy, residence time, and microstructure.

- The authors should clarify whether horizontal motion is negligible at these sites and, if not, how it may affect the comparison between PICO2303 and AHMIF.

- The velocity of the ice shield is not described.
* * *
**Specific and technical comments**

- **L 23**:

Replace *"including grain settling"* with *"structural compaction"*.

- **L 24**:

The statement that metamorphism generates a decrease in porosity is problematic. Metamorphism alone does not necessarily reduce porosity. Please revise and provide appropriate citations.

- **L 37 ff**:

The Clausius–Clapeyron relation is invoked without explicit references. Please add relevant citations and clarify how it is applied here.

- **L 40 / Figure 1**:

The derivation of the blue CC line is unclear. Accumulation rate is not equivalent to vapor pressure.

  - Is a secondary y-axis implicitly assumed?

  - Please clarify the physical meaning and construction of this line.

- **Spatial variability of accumulation**:

Please clarify whether spatial variability in accumulation within and between the sites has been quantitatively considered.

- **L 78 ff**:

Depth hoar formation is not "suppressed" at high densities; rather, a different type of depth hoar develops. Please revise and cite:

Pfeffer, W. T., and R. Mrugala (2002), *Journal of Glaciology*, 48(163), 485–494.

- **L 92, L 95**:

Please add degree symbols (°) to longitude and latitude coordinates.
* * *
**Recommendation**

The manuscript contains a valuable and rare dataset that is highly relevant to the Cryosphere community. However, **substantial revisions are required** to strengthen the quantitative analysis, clarify the physical interpretation, and support the proposed hypotheses. I recommend **major revision**.

Martin Schneebeli